# Gut Dysbiosis: A New Avenue for Stroke Prevention and Therapeutics

**DOI:** 10.3390/biomedicines11092352

**Published:** 2023-08-23

**Authors:** Shin Young Park, Sang Pyung Lee, Dongin Kim, Woo Jin Kim

**Affiliations:** 1Department of Clinical Laboratory Science, Cheju Halla University, 38 Halladaehak-ro, Jeju-si 63092, Republic of Korea; kj901217@gmail.com; 2Department of Neurosurgery, Brain-Neuro Center, Cheju Halla General Hospital, 65 Doryeong-ro, Jeju-si 63127, Republic of Korea; nsdr745@gmail.com; 3Department of Laboratory Medicine, EONE Laboratories, 291 Harmony-ro, Incheon 22014, Republic of Korea; gtphrase@eonelab.co.kr

**Keywords:** stroke, gut, dysbiosis, nutrition, inflammation, microbiota, mortality, therapeutics, blood, brain

## Abstract

A stroke is a serious life-threatening condition and a leading cause of death and disability that happens when the blood vessels to part of the brain are blocked or burst. While major advances in the understanding of the ischemic cascade in stroke was made over several decades, limited therapeutic options and high mortality and disability have caused researchers to extend the focus toward peripheral changes beyond brain. The largest proportion of microbes in human body reside in the gut and the interaction between host and microbiota in health and disease is well known. Our study aimed to explore the gut microbiota in patients with stroke with comparison to control group. Fecal samples were obtained from 51 subjects: 25 stroke patients (18 hemorrhagic, 7 ischemic) and 26 healthy control subjects. The variable region V3–V4 of the 16S rRNA gene was sequenced using the Illumina MiSeq platform. PICRUSt2 was used for prediction of metagenomics functions. Our results show taxonomic dysbiosis in stroke patients in parallel with functional dysbiosis. Here, we show that stroke patients have (1) increased *Parabacteroides* and *Escherichia_Shigella*, but decreased *Prevotella* and *Fecalibacterium;* (2) higher transposase and peptide/nickel transport system substrate-binding protein, but lower RNA polymerase sigma-70 factor and methyl-accepting chemotaxis protein, which are suggestive of malnutrition. Nutrients are essential regulators of both host and microbial physiology and function as key coordinators of host–microbe interactions. Manipulation of nutrition is expected to alleviate gut dysbiosis and prognosis and improve disability and mortality in the management of stroke.

## 1. Introduction

A stroke is a serious life-threatening condition and a leading cause of death and disability that happens when the blood vessels to part of the brain are blocked or burst. Stroke has a long history but the research of its pathobiology has only been carried out recently. Stroke is a kind of inflammation, and upon neurological injury, inflammatory cascades are initiated and subsequent recovery follows. Initial ischemic event results in excitotoxicity and oxidative stress which leads to microglial and astrocyte activation to secrete cytokines, glial fibrillary acidic protein (GFAP), and matrix metalloproteinases (MMPs). These factors are proinflammatory and lead to upregulated expression of cell adhesion molecules such as selectins and intracellular adhesion molecule 1 (ICAM-1) on endothelial cells, and recruit blood-derived inflammatory cells including neutrophils, macrophages, and lymphocytes to the ischemic site. In addition, dying neuronal cells release danger-associated molecular patterns (DAMPs) which in turn stimulate microglia and peripheral immune cells such as neutrophils, macrophages, and lymphocytes, causing production of proinflammatory factors, resulting in further activation of microglia and astrocyte [1].

While major advances in the understanding of the ischemic cascade in stroke were made over several decades, limited therapeutic options and high mortality and disability have caused researchers to extend their focus toward peripheral changes beyond the brain [2].

Various research, including animal model studies and clinical trials, highlighted the bidirectional connection between the brain and the gut in the pathogenesis of neurological and neuropsychiatric diseases such as depressive diseases, anxiety, bipolar disorder, autism, schizophrenia, Parkinson’s disease, Alzheimer’s disease, dementia, multiple sclerosis, and epilepsy [3,4,5,6,7,8,9,10,11,12]. The brain–gut axis comprises brain, spinal cord, autonomic nervous system (ANS), enteric nervous system (ENS), and hypothalamic–pituitary–adrenal (HPA) axis [13]. The vagus nerve (tenth cranial nerve) represents the principal pathway from the gut lumen to the brain, which acts collaboratively with several neurotransmitters released from ENS and has immunomodulatory properties [14]. The integrated activity of HPA and the vagus nerve permits high-level communication between the brain and gastrointestinal tissues, in more detail enterochromaffin cells, interstitial cells of Cajal, smooth muscle cells, enteric neurons, epithelial cells, and immune cells, that in practice are all regulated by the gut microbiota [15]. The largest proportion of microbes in the human body reside in the gut, and balanced intestinal microbiota stimulate a regulatory environment in the gut-associated lymphoid tissue (GALT) through the production and release of various immunomodulatory compounds, like short-chain fatty acids (SCFAs) [16], but any alteration in gut microbiota and host communication can be considered as a triggering element in the pathogenesis of diseases [17].

A symbiotic relationship with gut microbiota is an essential component to maintain host heath both metabolically and immunologically. The host utilizes the gut microbiota to digest food to obtain the nutrients and energy and participates in host metabolism and organism immunity [18]. One-third of the small molecules in the blood derived from gut microbiota and the trillions of commensal or mutualistic bacteria and archaea are a huge chemical factory that can produce some essential amino acids and vitamins and many compounds that affect host energy homeostasis, body adiposity, glucose tolerance, insulin sensitivity, inflammation, and hormone secretion needed for their own existence and survival [19,20]. Fiber fermentation by fecal microbial flora produces the short-chain fatty acids (SCFAs) such as butyrate, propionate, and acetate, which affect host metabolism in various ways by acting on G protein-coupled receptors (GPCRs) expressed by enteroendocrine cells. Butyrate and acetate activate glucagon-like peptide 1 (GLP-1) and peptide YY (PYY) secretion with influence on GLP-1-induced insulin biosynthesis in the pancreas and on PYY-induced satiety in the brain, and acetate may upregulate fat storage by inducing release of ghrelin [21]. Microbiota-derived succinate stimulates expression of uncoupling protein 1 (UCP1), leading to upregulate thermogenesis in adipose tissue [22]. Primary bile acids are transformed by gut microbiota to secondary bile acids which act through the Takeda-G-protein-receptor-5 (TGR5) receptor to upregulate GLP-1 secretion causing thermogenesis in adipose tissue [23]. Indole and its derivatives are the ligand to the aryl hydrocarbon receptor (AhR). Indole-3-propionic acid is associated with enhanced insulin secretion and insulin sensitivity and thereby results in a reduced risk of type 2 diabetes [24]. Commensal bacteria produce N-acyl amide, a mimic of human signaling molecules, which controls host glucose metabolism by binding to the G protein-coupled receptor 119 (GPR119) [25]. Proteins released by gut symbionts also regulate paracrine or endocrine action. Caseinolytic peptidase B (ClpB), a protein released by *Escherichia coli*, is implicated in the control of appetite [26]. Melanocortin-like peptide of *E. coli* (MECO-1), which is a structural mimic to α-melanocyte stimulating hormone and adrenocorticotropin, functions through the mammalian melanocortin-1 receptor (MC1R), suppressing cytokine secretion in response to proinflammatory stimulation [27]. Amuc_1100, expressed on the outer membrane of *Akkermansia muciniphila*, ameliorates gut barrier function with elevated goblet cell population via Toll-like receptor 2 (TLR2) and shows the beneficial effect on insulin sensitivity and energy metabolism [28]. Host metabolism is also affected by synthesized neurotransmitters by gut commensals such as catecholamine, histamine, γ-aminobutyric acid and serotonin or gaseous neurotransmitters, including nitric oxide (NO) and hydrogen sulfide (H2S) [29].

The immune system orchestrates the maintenance of key features of host–microbe symbiosis while the microbiome plays critical roles in the training and development of major components of the host’s immune system [30]. Trace metals are essential micronutrients required for survival across all kingdoms of life, and hosts have numerous strategies of metal limitation and intoxication that prevent bacterial proliferation, a process termed nutritional immunity [31]. Fecal calprotectin allows the chelation of essential divalent metal ions (e.g., calcium, iron, or zinc), limiting growth of invasive and commensal gut bacteria, while it represents a well-studied inflammatory biomarker in inflammatory bowel diseases [32].

We have previously measured fecal calprotectin (FC), a gut nutritional immunity marker of host, in stroke and reported two meaningful findings: (1) there was a significant increase in FC levels in stroke patients compared to those in controls; (2) FC in stroke patients had negative correlation with levels of albumin and lymphocyte but positive correlation with C-reactive protein (CRP) [33]. This study aimed further to explore the gut microbiota, the other side of host and microbe interaction, in stroke patients with comparison to control group.

## 2. Materials and Methods

Fecal samples were obtained from 51 subjects: 25 stroke patients (18 hemorrhagic, 7 ischemic) and 26 healthy control subjects. The variable region V3–V4 of the 16S rRNA gene was sequenced using the Illumina MiSeq platform. PICRUSt2 was used for prediction of metagenomics functions.

### 2.1. Subjects

In the present study, 51 subjects provided a single fecal sample including 25 stroke patients (STR) and 26 healthy control subjects (CON). Stroke patients, including both ischemic and hemorrhagic types, were directly admitted from the emergency department (ER) to the intensive care unit (ICU). Fecal samples were obtained from stroke patients as early as possible during their stay in the ICU. The enrolled period of the subjects was from September 2018 to April 2019. Informed consent was written and obtained from the subjects or their guardians. Diagnosis of stroke was based on head computed tomography (CT) scans or magnetic resonance imaging (MRI) studies. The study subjects had no history of colorectal or systemic inflammatory conditions. The present study was approved by the ethical review board of Cheju Halla University (IRB approval number: 1044348-20180713-HR-007-01).

### 2.2. Demographic Characteristics

Baseline demographic and clinical characteristics of the stroke patients were collected (Table 1). These included patient sex, age, body mass index (BMI), stroke type, and comorbidities. The age of the stroke patients ranged from 39 to 86 and the age of the control subjects ranged from 41 to 85. The male-to-female ratio of the stroke patients was 14:11 and the male-to-female ratio of the control subjects was 16:10. The body mass index (BMI) of the stroke patients ranged from 17.3 to 31.4 and the BMI of the control subjects ranged from 18.8 to 33.2. Comorbidities in stroke patients include diabetes mellitus (16%), hypertension (52%), and coronary artery disease (8%), while those in control subjects include diabetes mellitus (15.3%) and hypertension (30.7%). Regarding medication, intravenous antibiotics were taken along the progress in all stroke patients. While stroke was reported to be associated with obesity, hypertension, or diabetes mellitus, all of which had major effect on gut dysbiosis [34], the subject difference between stroke patients and control group was not as significant for those comorbidities.

### 2.3. DNA Extraction

Subjects provided a single fecal sample for measurement as early as possible during their ICU stay. Aliquots of fecal samples were kept frozen on receipt at −80 °C prior to DNA extraction [35]. DNA was extracted from 200 mg of fecal aliquot using a QIAamp PowerFecal Pro DNA Kit (Qiagen, Hilden, Germany) according to the manufacturer’s instructions without modification. Briefly, the fecal sample and eight hundred microliters of lysis buffer (solution CD1) were applied to the PowerBead Pro tube. The mixture was horizontally agitated at maximum speed for 10 min until the fecal sample became homogeneous. The homogenate was then centrifuged at 15,000× *g* for one minute. The supernatant was removed and mixed with two hundred microliters of solution CD2, which contained inhibitor removal reagents. After the centrifuge stopped, the supernatant was removed and mixed with six hundred microliters of Solution CD3, which customized the DNA solution salt concentration and allowed more specific deoxynucleic acid binding to the column. The mixture was then carefully added to the QIAamp spin column and centrifuged at maximum speed for one minute. The column was cleaned two times with washing buffers (solutions EA and C5, respectively). Then, a total of one hundred microliters of elution buffer (Solution C6) was applied and centrifuged at maximum speed for one minute to elute the deoxynucleic acid [36].

### 2.4. Library Construction and Sequencing

The sequencing libraries were prepared according to the Illumina 16S Metagenomic Sequencing Library protocols to amplify the V3 and V4 regions. The 4 ng of input gDNA was polymerase chain reaction (PCR) amplified with 2× KAPA HiFi HotStart ReadyMix buffer (KAPA Biosystems, Wilmington, MA, USA), one micromole each of universal F/R PCR primers. The protocols for PCR were as follows: (1) heat activation for three minutes at 95 °C; (2) twenty five cycles of thirty seconds at 95 °C, thirty seconds at 55 °C, and thirty seconds at 72 °C; (3) five minutes final extension at 72 °C. The universal primer pairs with Illumina adapter overhang sequences were: V3-F: 5′-TCG TCG GCA GCG TCA GAT GTG TAT AAG AGA CAG CCT ACG GGN GGC WGC AG-3′, V4-R: 5′- GTC TCG TGG GCT CGG AGA TGT GTA TAA GAG ACA GGA CTA CHV GGG TAT CTA ATC C-3′. The PCR amplicon was cleaned with AMPure beads (Agencourt Bioscience, Beverly, MA, USA). After purification, five microliters of the PCR product was PCR amplified the second time for the last library construction holding the index using NexteraXT Indexed Primer. The protocol condition for the second PCR was equal to that for the first PCR except for eight cycles. The PCR amplicon was cleaned with AMPure beads. The last purified product was then quantitatively measured using a Qubit (Life Technologies, Carlsbad, CA, USA) 2.0 fluorometer, following the manufacturer’s instructions, and qualified using the TapeStation D1000 ScreenTape (Agilent Technologies, Waldbronn, Germany). The paired-end (2 × 300 bp) sequencing was performed using the MiSeq™ platform (Illumina, San Diego, CA, USA) [37].

### 2.5. Sequencing Data Analysis

Forward and reverse paired-end 16S rRNA sequences were combined by Quantitative Insights Into Microbial Ecology (QIIME) 2 (version: 2021.2) pipeline. After merging, the sequences were demultiplexed and separated into samples using the index sequence of each sample. With the use of QIIME 2 plugin DADA2, a quality check was carried out, the noise was removed, and denoised and filtered amplicon sequencing variants were rarefied to a depth of 4000 sequences per sample. Analysis of alpha diversity was carried out by the diversity plugin of QIIME 2. Nonmetric multidimensional scaling (NMDS) diagrams were produced by R packages “phyloseq” and “ggplot2”. Taxonomy was designated to the amplicon sequence variants (ASVs) by Vsearch pretrained on Silva reference database (silva-138-99-seqs) and the feature-classifier plugin of QIIME 2 [38]. The compositional microbiome data and their relative abundances were calculated on seven levels of taxonomy including species, genus, family, order, class, phylum, and kingdom. The linear discriminant analysis (LDA) effect size (LEfSe) method was carried out to reveal the differences in taxon abundance between groups at different taxonomic levels using default parameters except otherwise specified [39]. Two groups were regarded as significantly different at a *p* value less than 0.05 and a |log10(LDA score)| more than 2. The function prediction analysis was carried out using the Phylogenetic Investigation of Communities by Reconstruction of Unobserved States (PICRUSt) software based on Kyoto Encyclopedia of Genes and Genomes (KEGG) database. PICRUSt2 is a software for predicting functional abundances based only on marker gene sequences such as 16S rRNA gene sequencing data, and function refers to gene families such as KEGG orthologs [40].

### 2.6. Statistical Analysis

For each group, twenty-five participants were necessary to have a power of ninety percent with two-sided five percent significance and 0.2 for an effective size, according to Whitehead et al. [41]. The Student’s *t*-test or Fisher’s exact test were utilized according to variables to compare demographic characteristics of the study subjects. All tests were two-sided, and *p* < 0.05 was considered statistically significant. The statistical software package for data was the GraphPad Prism version 10.0.2 (GraphPad Software Inc., La Jolla, CA, USA). Beta diversity was measured from the pairwise PERMANOVA (permutational multivariate analysis of variance) test using Bray–Curtis and Jaccard distance metrices [42].

## 3. Results

### 3.1. Alpha Diversity

The Student’s *t*-test indicated differences in the richness and evenness of the bacteria between two groups (Figure 1). The bacteria were less enriched in the STR group (*p* = 0.01) (Figure 2A). There was also a difference in the evenness between two groups (*p* = 0.001) (Figure 2B). To sum up, the gut microbiome was less abundant in the STR group.

### 3.2. Beta Diversity

When observed at the genus level, control and stroke groups differed significantly in bacterial diversity, while there was no difference at phylum level (Figure 2). In Figure 2, Bray–Curtis (A,C) and Jaccard (B,D) distance metrics were used. Consistent results were obtained in both metrics.

### 3.3. Taxonomy Changes

We performed LEfSe to identify bacterial taxa that could best explain the differences between CON and STR with alpha value for the factorial Kruskal–Wallis test among classes = 0.05, alpha value for the pairwise Wilcoxon test between subclasses = 0.05, and threshold on the logarithmic LDA score for discriminative features > 3.6. The clades that decreased in stroke patients included Prevotella, Fecalibacterium, Roseburia, Lachnospiraceae_NK4A136_group, Eubacterium, and Dialister while the clades that increased in stroke patients included Parabacteroides, Lachnoclostridium, Escherichia_Shigella, Fusobacterium, Lactobacillales, and Enterococcus (Figure 3).

### 3.4. Biologic Function Changes

We performed LEfSe to identify biologic functions that could best explain the differences between CON and STR with alpha value for the factorial Kruskal–Wallis test among classes = 0.05, alpha value for the pairwise Wilcoxon test between subclasses = 0.05, and threshold on the logarithmic LDA score for discriminative features > 2.0 (Figure 4).

The functions that were decreased in stroke patients included RNA polymerase sigma-70 factor, putative ABC transporter system permease, methyl-accepting chemotaxis protein, glutamate synthase, thiamin-phosphate pyrophosphorylase, ATP-dependent DNA helicase RecG, putative hydrolase of the HAD superfamily, phosphoglycolate phosphatase, and GTP pyrophosphokinase.

The functions that were enriched included transposase, peptide/nickel transport system substrate-binding protein, cysteine-S-conjugate beta-lyase, peptide/nickel transport system ATP-binding protein, peptide/nickel transport system permease protein, ribose transport system ATP-binding protein, alpha-glucosidase, AraC family transcriptional regulator, iron (III) transport system permease protein, iron (III) transport system substrate binding protein, GntR family transcriptional regulator, iron complex transport system permease protein, putative tricarboxylic transport membrane, and dihydrolipoyl dehydrogenase [43]. Taken together, our results showed taxonomic dysbiosis in stroke patients in parallel with functional dysbiosis.

## 4. Discussion

While stroke is a vascular event and subsequent inflammation which occurs in the brain, the dysfunction of the gut–brain axis has been known to be a hopeful area of research for identifying preventive and treatment strategies against stroke [2]. Stroke is characterized by a disruption of blood supply to a specific region of the brain, leading to neuronal damage and death as well as disturbances in the blood–brain barrier and is complicated by functional deficits of motor, sensing, or cognition and seizures or depression [44]. Elaboration on the neural substrates affected by stroke can shed light on the potential mechanisms underlying the interaction between the gut and the brain, like in migraine or neuropathic pain, where such neural substrates or pathomechanisms which involve calcitonin gene-related peptide (CGRP), transient receptor potential channels (TRP channels), endocannabinoid system, glutamatergic system, tryptophan-kynurenin (Trp-KYN) metabolism, neuroinflammation, cytokines, and microglial activation were revealed [45]. In recent years, several efforts have been made and emerging evidence suggests that ischemic brain tissue and activated microglia release cytokines and damage-associated molecular patterns (DAMPs), which leads to triggering vascular endothelial cells to reveal adhesion molecules and to extravasate immune cells and inflammatory cells to the injury site of stroke from the blood circulation. In the meantime, production of cytokines and DAMPs in addition to stimulation of the vagus nerve, which acts as the primary communication pathway between the central nervous system (CNS) and enteric nervous system (ENS), lead to increase in intestinal permeability, gut dysbiosis, and gut dysmotility, resulting in bacterial translocation in intestine and migration of gut immune cells and inflammatory cells into the injury sites of stroke through circulating blood flow. Yet, the exact molecular landscape which underlies the alterations in the brain–gut axis is in its infancy [46,47].

We collected fecal samples and analyzed composition and KEGG function of gut microbiota both from patients with stroke and health controls. Our results showed (1) decreased alpha diversity, (2) different microbiomes from control subjects at genus-level beta-diversity, (3) dysbiotic change in bacterial abundances, and (4) dysbiotic change in biologic functions in stroke patients.

Peh et al. have recently reviewed 14 clinical human stroke studies globally and found that the main cohort background was Chinese (12/14). They revealed that alpha diversity in stroke was heterogeneous (no difference: 5, decreased: 4, increased: 3, N/A: 2) and 62 upregulated (e.g., *Streptococcus*, *Lactobacillus*, *Escherichia*) and 29 downregulated (e.g., *Eubacterium*, *Roseburia*) microbial taxa in stroke patients [48]. Our study showed that the patients group for the gut showed enrichment of *Parabacteroides*, Tannerellaceae, *Lachnoclostridium*, *Escherichia_Shigella*, *Fusobacterium*, *Clostridia_UCG_014*, and *Enterococcus* in contrast to depletion of *Prevotella*, *Faecalibacterium*, *Roseburia*, Selenomonadales, *Lachnospiraceae_NK4A136_group*, *Eubacterium*, and *Dialister*. Among the aforementioned 14 clinical studies, two studies had more than 300 in cohort size and our results are consistent with their results. Yin et al. studied 322 Chinese stroke patients and 231 controls and showed that Proteobacteria was enriched but *Bacteroides*, *Prevotella*, and *Faecalibacterium* were depleted [49]. Haak et al. studied 349 Dutch stroke patients and 51 controls and showed that Proteobacteria, *Escherichia/Shigella*, *Peptoniphilus*, *Ezakiella*, and *Enterococcus* were enriched while Firmicutes and Bacteroidetes were depleted [50].

There are few reports about the biologic function of gut microbiota in stroke. Sun et al. studied 132 Chinese stroke patients and divided them into good (*n* = 105) and poor (*n* = 27) outcome based on a 3-month modified Rankin scale. The functional potential was predicted using the Phylogenetic Investigation of Communities by Reconstruction of Unobserved States (PICRUSt), and showed that upregulated function in the poor outcome group included membrane transport, transcription, and metabolism while downregulated functions included amino acid metabolism, metabolism of cofactors and vitamins, and replication and repair [51]. These findings are consistent with our results. Our study showed that membrane transport (ribose transport system), transcription (AraC family transcription regulator), and metabolism (cysteine-S-conjugate beta-lyase) are enriched in stroke patients while amino acid metabolism (glutamate synthase), replication and repair (ATP-dependent DNA helicase RecG), and metabolism of cofactors and vitamins (thiamine-phosphate pyrophosphorylase) are depleted in stroke patients.

We have previously collected blood samples from stroke patients and analyzed various parameters; a notable finding is low levels of albumin and lymphocytes [33]. Albumin and lymphocytes serve as markers of prognostic nutritional index (PNI), and depletion of both markers is associated with poor functional outcome [52]. Stroke is regularly accompanied by dysphagia and other factors associated with decreased nutritional intake [53]. Nutrients are essential regulators of both host and microbial physiology and function as key coordinators of host–microbe interactions [54]. Trace metals are important nutrients for all forms of life. Murdoch et al. recently reviewed nutritional immunity for nutrient metals at the host–microbe interface [31], and oral cuprizone, copper chelator, was reported to induce a demyelination model simulating progressive multiple sclerosis and suppress the tryptophan-kynurenine metabolic system [55].

We have reported increase in stroke patients of fecal calprotectin [33], which binds to and sequesters Zn, Mn, Ni, Cu, and Fe in the extracellular milieu through the action of metal-binding sites [56], and our present study showed that gut microbiome in stroke patients became enriched in the nickel transport system and the iron transport system.

In this regard, malnutrition can increase bacterial infections and systemic inflammation that reversibly impact brain tissue degeneration. Recent preclinical studies have shown the protective effects of nutritional support. The administration of a high-protein (HP) diet in rats has been shown to reduce post-stroke neurological deficit [57], and a HP diet in radiation-induced acute-phase inflammation in rats led to increased percentage of lymphocytes and decreased percentage of neutrophils [58].

As the prognostic importance of nutrition led three German medical societies to publish the guidelines of clinical nutrition in patients with stroke [59] in 2013, the manipulation of diet contents for stroke management, understanding the specific roles and underlying mechanisms of nutrients in regulating the host–microbe interactions, and development of strategies for improving prognosis of stroke have been gaining attention.

The potential to modulate the activity of the immune system by interventions with specific nutrients is termed “immunonutrition”, and this concept may be applied to stroke [60]. Immunonutrition has been reported to improve wound healing and reduce infectious complications and length of stay in hospital. Its formulation includes supplementation with arginine, glutamine, omega-3 fatty acids, vitamins, and trace minerals (zinc, selenium), some of which have commonly been classified as nonessential but have become essential in certain clinical situations, such as for trauma patient or patients at high risk for malnutrition [61]. Arginine has various effects on wound healing and immune function. Metabolically, arginine is a precursor for ornithine, which is essential for both polyamine synthesis and NO. It is also a precursor to proline, and is thus engaged specifically for collagen synthesis [62]. Recent studies in rodents and humans showed that supplemental arginine-induced gut microbiota remodeling with enrichment of *B. pseudolongum* boosts pulmonary immune defense against nontuberculous mycobacteria (NTM) infection by driving the protective gut–lung axis in vivo [63], and arginine treatment decreases neuronal death after rat cerebral ischemia/reperfusion (I/R) injury and improves functional recovery of stroke animals [64]. Glutamine is normally nonessential but has become “conditionally essential” during inflammatory conditions. Glutamine is important to cell proliferation in that it can act as a respiratory fuel and that it can enhance the function of stimulated immune cells [65]. Glutamine protects mouse brain from ischemic injury via upregulating heat shock protein 70 [66], and reduces the intestinal colonization and bacterial overgrowth or bacterial translocation [67]. Essential fatty acids play an important role in the immune system by regulating properties of cell membranes and controlling cell signaling, while Omega-3 fatty acids lessen inflammatory responses through their effects on production of specific chemokines and cytokines [68]. Omega-3 polyunsaturated fatty acids enhance cerebral angiogenesis and provide long-term protection after stroke [69], and they correlate with gut microbiome diversity [70]. Vitamin A is an essential micronutrient that comes in multiple forms, including retinols, retinals, and retinoic acids (RAs). It plays a role in the inflammatory phase of wound healing and has been demonstrated to enhance production of extracellular matrix components such as collagen type I and fibronectin [71]. Administration of a combination of vitamin A and D supplementation can significantly increase vitamin A and D serum levels, decrease IL-1β serum levels, and ultimately improve clinical outcome in ischemic stroke patients [72]. The gut flora was altered by a vitamin-A-deficient diet in rats and mice, and RA could restore *Lactobacilli* that were downregulated in a murine lupus model [73]. Vitamin C plays an essential role in collagen formation and post-translational modification, and its deficiency leads to scurvy with various cutaneous and wound manifestations. It acts as a cofactor in the hydroxylation of proline and lysine residues in procollagen, which are critical for the stability of collagen fibers. In addition, vitamin C enhances neutrophil motility [74]. Supplementation with vitamin C increased the abundance of bacteria of the genus *Bifidobacterium* [75], and post-stroke treatment with high-dose ascorbate protects the brain through epigenetic reprogramming and may function as a robust therapeutic against stroke injury [76]. Zinc is a cofactor in a number of intracellular enzymatic reactions pertaining to wound healing. It is also an antioxidant and confers resistance against epithelial apoptosis [77]. Zinc improves neurological recovery by promoting angiogenesis via the astrocyte-mediated HIF-1α/VEGF signaling pathway in experimental stroke [78]. Selenium has a strong antioxidant role, and organic selenium was associated with a higher concentration of total VFA, propionate, and butyrate, a higher number of DNA copies of *Lactobacillus*, and a trend to lower DNA copies of *Escherichia coli* [79]. Lower selenium levels were associated with worse stroke outcomes, and selenase improved the modified Rankin Scale and National Institute of Health Stroke Scale scores significantly [80].

Limitations of this study include relatively small sample size and predictive biologic functions based on PICRUSt2. Though our taxonomic result is consistent with a large cohort size of more than 300 Chinese and Dutch people and predictive biologic functions are consistent with a Chinese prognosis study, future directions include a larger sample size of Korean cohort and whole-genome sequencing-based biologic functions and metabolomics study.

## 5. Conclusions

Knowledge of the gut–brain axis and metabolic and immunological interaction between host and microbe is shedding light on the research of stroke with limited therapeutic options in spite of high mortality and disability. We previously reported that fecal calprotectin had an association with the Glasgow Coma Scale, which is suggestive of gut–brain axis and deficiency of blood albumin and lymphocytes in stroke patients. This time, we explored the gut microbiome, the other side of host and microbe interaction, through 16S rRNA sequencing and found taxonomic dysbiosis in stroke patients in parallel with functional dysbiosis, which is suggestive of malnutrition. Stroke is commonly accompanied by dysphagia and other gastrointestinal complications associated with decreased nutritional intake, and the nutritional status is correlated with prognosis of stroke patients. In contrast to symbiotic interaction, dysbiotic microbiome cannot provide beneficial metabolism to the host. Nutrients are common denominators and essential regulators of both host and microbial physiology and function as key coordinators of host–microbe interactions. The potential to modulate the activity of the immune system by interventions with specific nutrients is termed “immunonutrition”, and this promising concept may be applied to enhanced management of stroke with regard to our current study.

## Figures and Tables

**Figure 1 biomedicines-11-02352-f001:**
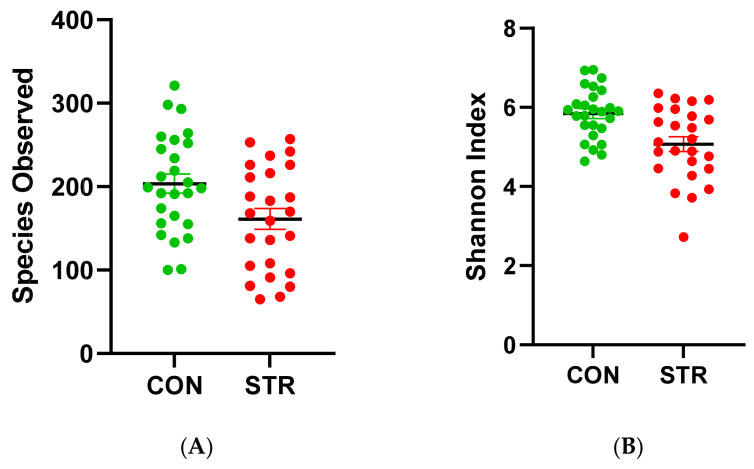
(**A**) Observed amplicon sequence variants (ASVs); *p* = 0.01. (**B**) Shannon index; *p* = 0.001; CON: control; STR: stroke; CON (green): control; STR (red): stroke.

**Figure 2 biomedicines-11-02352-f002:**
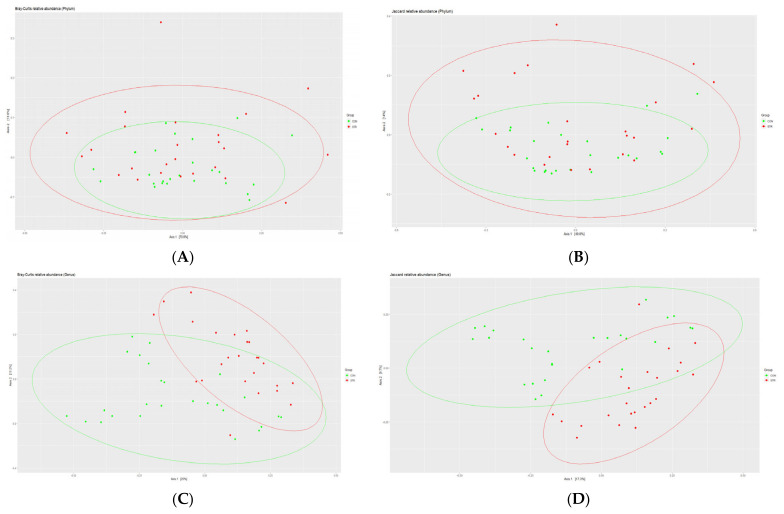
Beta diversity observation in two groups. (**A**) Bray–Curtis plot of control and stroke group microbiome communities at phylum level; *p* = 0.354. (**B**) Jaccard plot at phylum level; *p* = 0.165. (**C**) Bray–Curtis plot at genus level; *p* = 0.001. (**D**) Jaccard plot at genus level; *p* = 0.001; CON (green): control; STR (red): stroke.

**Figure 3 biomedicines-11-02352-f003:**
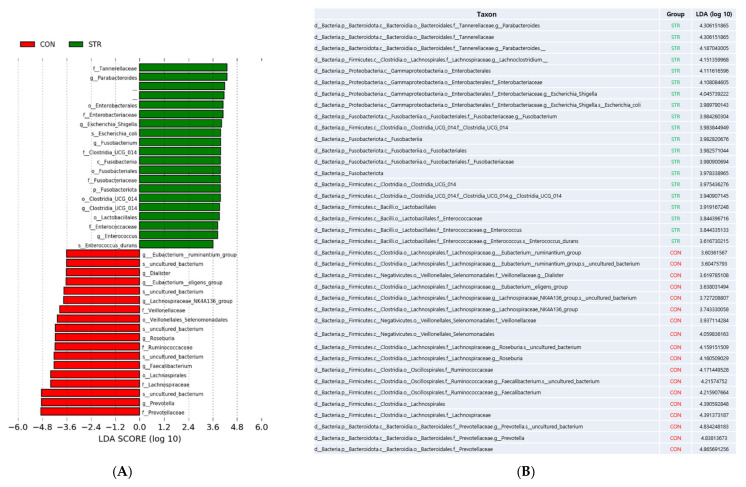
(**A**) Linear discriminant analysis effect size (LEfse) result showing taxonomic difference between two groups. (**B**) Taxon table and LDA score of each taxon; CON (red): control; STR (green): stroke.

**Figure 4 biomedicines-11-02352-f004:**
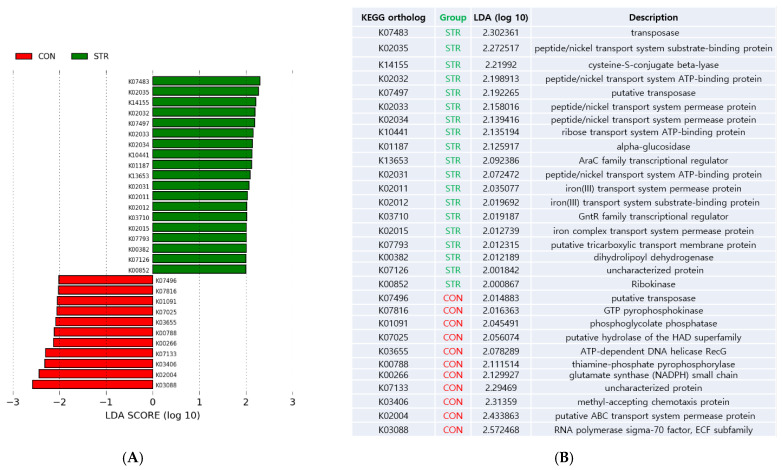
(**A**) Linear discriminant analysis effect size (LEfse) result showing functional difference between two groups. (**B**) KEGG ortholog table and LDA score of each ortholog; CON (red): control; STR (green): stroke.

**Table 1 biomedicines-11-02352-t001:** Demographic characteristics of the study population.

	CON (*n* = 26)	STR (*n* = 25)	*p*
Age (year)	58.6 ± 14.1	60.8 ± 15.2	0.579
Sex (M/F)	16/10	14/11	0.663
BMI (kg/m^2^)	23.4 ± 3.5	23.7 ± 3.4	0.808
Comorbidities			
DM (number)	4	4	1
Hypertension (number)	8	13	0.159
CAD (number)	0	2	0.235
Medication			
PPI (number)	0	0	NA
NSAID (number)	0	0	NA
ABX (number)	0	25	<0.0001

Abbreviations: CON, healthy controls; STR, stroke patients; BMI, body mass index; DM, diabetes mellitus; CAD, coronary heart disease; PPI, proton pump inhibitor; NSAID, nonsteroidal anti-inflammatory drug; ABX, antibiotics; NA, not applicable. Values are expressed as mean ± SD.

## Data Availability

Data supporting the reported result can be accessed by corresponding with the authors.

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
