# Peer review of "Gut Dysbiosis: A New Avenue for Stroke Prevention and Therapeutics"

_biomedicines, 2023, doi:10.3390/biomedicines11092352_

Round 1

Reviewer 1 Report

Dear authors,

After the review process, I have several comments: the process of samples obtaining and preservation of the characteristics should be clearly described with references; the level of possible negative data should be included; you should recent contributions made to alleviating human dysbiosis in degenerative diseases, especially for neurodegenerative pathologies based on the rising prevalence of obesity; also, the balance of the gut–brain axis has a major significance in establishing homeostasis due to neurotransmitters; the metabolic response is an important detail because is mediated by bioactive compounds in the diet that regulate the synthesis of critical metabolites, such as SCFAs.

Best regards!

Author Response

We appreciate for your precious comments.

Following your instructions, we revised introduction and methods of the original manuscript. We are sorry that we could not fully grasp the meaning of "the level of possible negative data should be included" in your comments and could not respond properly.

Reviewer 2 Report

The paper entitled ‘Gut Microbiota Composition and Function in Stroke Patients: A Comparative Study’ investigates the gut microbiota composition and function in stroke patients compared to a control group. The authors aim to explore the potential role of the gut-brain axis in stroke and identify potential therapeutic strategies. In the introduction, the authors highlight the historical context of stroke research and the limited therapeutic options available. They emphasize the need to focus on peripheral changes beyond the brain and cite a recent study by Park et al. that found increased fecal calprotectin levels in stroke patients. The materials and methods section describes the study subjects, sample collection, and DNA extraction process. Fecal samples were obtained from 25 stroke patients and 26 healthy control subjects. DNA extraction was performed using a commercial kit, and sequencing libraries were prepared for 16S rRNA analysis. The results section presents the findings of the study. The alpha diversity analysis revealed decreased bacterial richness and evenness in the gut microbiota of stroke patients compared to controls. Beta diversity analysis showed significant differences in bacterial diversity at the genus level. The study identified specific bacterial taxa that were enriched or depleted in stroke patients compared to controls. The authors conducted functional analysis using the PICRUSt software based on the KEGG database. They found dysbiotic changes in biologic functions in stroke patients, including alterations in membrane transport, transcription, and metabolism. The discussion section interprets the findings in the context of previous studies on stroke and gut microbiota. The authors discuss the implications of gut dysbiosis in stroke and its potential association with malnutrition. They also highlight the need for further research to understand the underlying mechanisms and explore dietary interventions for stroke management.

In general, I think the idea of this manuscript is really interesting and the authors’ fascinating observations on this timely topic may be of interest to the readers of Biomedicines. However, some comments, as well as some crucial evidence that should be included to support the author’s argumentation, needed to be addressed to improve the quality of the manuscript, its adequacy, and its readability prior to the publication in the present form, in particular reshaping parts of the Introduction and Methods sections by adding more evidence and theoretical constructs.

Please consider the following comments:

According to the Journal’s guidelines, Authors' full first and last names must be provided, as well as their academic affiliations.

Abstract: Authors should consider rephrasing this section. According to the Journal’s guidelines, the Abstract should contain most of the following kinds of information in brief form. Please, consider giving a more synthetic overview of the paper's key points: I would suggest rephrasing the results and conclusion to make them clear for readers to understand.

Introduction: This section would benefit from providing more context and background information on stroke and its pathobiology. Furthermore, while this section sets the stage for the study, it would benefit from incorporating a discussion on the neural substrates of stroke to enhance the understanding of the brain-gut axis and its relevance to stroke pathophysiology. In this regard, to provide a more comprehensive background, it would be valuable to include a discussion on the neural substrates involved in stroke. Stroke is characterized by a disruption of blood supply to a specific region of the brain, resulting in neuronal damage and functional deficits (https://doi.org/10.1016/j.neubiorev.2023.105163). Elucidating the neural substrates affected by stroke can shed light on the potential mechanisms underlying the interaction between the gut and the brain. Studies have shown that stroke impacts multiple brain regions involved in motor, sensory, and cognitive functions, including the cortex, basal ganglia, thalamus, and brainstem (DOI: 10.3390/biomedicines11030945; https://doi.org/10.17219/acem/166476). These neural substrates play crucial roles in the regulation of physiological processes, such as autonomic control, neurotransmitter release, and neuroinflammatory responses. Therefore, investigating the gut microbiota in the context of stroke provides an opportunity to explore how alterations in the neural substrates of stroke may contribute to gut dysbiosis and vice versa.

Materials and Methods: In my opinion, the description of the subjects should include more details about their characteristics, such as age, gender, and any relevant clinical information. Also, consider describe the DNA extraction more in detail, including specific steps and reagents used, as well as information about the sequencing platform used and any quality control measures taken during the sequencing process.

Discussion: The discussion section needs to provide a more comprehensive interpretation and analysis of the results. Here, the Authors should better discuss the implications of the findings in relation to the existing knowledge in the field. I think the ‘Conclusions’ paragraph would benefit from some thoughtful as well as in-depth considerations by the authors, because as it stands, it lists down all the main findings of the research, without really stressing the theoretical significance of the study. Authors should make an effort, trying to explain the theoretical implication as well as the translational application of their research.

In my opinion, a proper ‘Conclusion’ paragraph in which summarize the key findings of this study is necessary, to properly explain the theoretical implication as well as the translational application of their research.

In according to the previous comment, I would ask the authors to include a proper and defined ‘Limitations and future directions’ section before the end of the manuscript, in which authors can describe in detail and report all the technical issues brought to the surface.

Regarding the Tables: Please provide an explanatory caption for each figure and table within the text. 

I hope that, after these careful revisions, the manuscript can meet the Journal’s high standards for publication. I am available for a new round of revision of this article. 

Best regards,

Reviewer

Minor editing of English language required.

Author Response

We appreciate for your precious comments.

Following your instructions, we revised all sections of original manuscript including limitations and future directions in discussion section and explantory caption of tables and figures within the text.

Round 2

Reviewer 1 Report

Dear authors,

 In comments on the results of demographic characteristics, you should include the relation between gut dysbiosis, obesity, and stroke/ It is an essential connection related to the rest of the comorbidities mentioned by you.

Best regards

Author Response

Dear Reviewer 1,

We appreciate for your precious comments.

Following your instruction on demographic characteristics, we revised the manuscript in material and method section (Highlighted blue).

Best regards,  

Reviewer 2 Report

Dear Authors,

I would like to express my appreciation for your insightful study on the brain-gut axis and its relevance to stroke pathophysiology. I found your approach fascinating and believe your findings have the potential to advance the field further. Nevertheless, I wanted to highlight an area where I believe additional information could enhance the comprehensiveness of your research. In particular, I recommend including a discussion on the neural substrates involved in stroke to provide a more holistic background for your readers. As you know, stroke is characterized by a disruption of blood supply to a specific region of the brain, leading to neuronal damage and functional deficits. It would be beneficial to elaborate on the neural substrates affected by stroke, as this can shed light on the potential mechanisms underlying the interaction between the gut and the brain. To support this suggestion, I still believe that it would be beneficial to add some evidence focusing on the role of how neural substrates are impacted by stroke and their roles in motor, sensory, and cognitive functions (DOI: 10.3390/ijms24044114; https://doi.org/10.1016/j.neubiorev.2023.105163). Integrating this information into your manuscript could significantly enrich your research, as it strengthens the basis for exploring the gut-brain axis in the context of stroke. Understanding how alterations in the neural substrates of stroke may contribute to gut dysbiosis and vice versa presents a valuable opportunity for further investigations. Thank you for considering these suggestions. Your dedication to advancing scientific knowledge is truly commendable, and I look forward to seeing your manuscript reach new heights with these additions.

Overall, this is a timely and needed work. It is well researched and nicely written, with a good balance between descriptive and narrative text. I just need to do one more comment, to provide a clearer and more understandable neural background to the readers, before continuing with the publication.

Best regards,

Reviewer

Author Response

Dear Reviewer 2,

We appreciate for your precious comments.

Following your instruction, we revised the manuscript in discussion section (Highlighted blue).

Best regards,